# Synthesis of Nano-Calcium Oxide from Waste Eggshell by Sol-Gel Method

**Lulit Habte [1]**, **Natnael Shiferaw [2]**, **Dure Mulatu [1]**, **Thriveni Thenepalli [3]**,
**Ramakrishna Chilakala [4] and Ji Whan Ahn [3,\*]**

[1]   Department of Resources Recycling, University of Science & Technology, 217 Gajeong-ro, Gajeong-dong, Yuseong-gu, Daejeon 34113, Korea; luna1991@ust.ac.kr (L.H.); dure@kigam.re.kr (D.M.)

[2]   Korea Research Institute of Climate Change, 11, Subyeongongwon-gil, Chuncheon-si 24239, Korea; natlulit@gmail.com

[3]   Center for Carbon Mineralization, Mineral Resources Division, Korea Institute of Geosciences and Mineral Resources (KIGAM), 124 Gwahagno, Gajeong-dong, Yuseong-gu, Daejeon 34132, Korea; thenepallit@rediffmail.com

[4]   Department of Bio-based Materials, School of Agriculture and Life Science, Chungnam National University, Daejeon City 34134, Korea; chilakala_ramakrishna@rediffmail.com

\*   Correspondence: ahnjw@kigam.re.kr

**Abstract:** The sol-gel technique has many advantages over the other mechanism for synthesizing metal oxide nanoparticles such as being simple, cheap and having low temperature and pressure. Utilization of waste materials as a precursor for synthesis makes the whole process cheaper, green and sustainable. Calcium Oxide nanoparticles have been synthesized from eggshell through the sol-gel method. Raw eggshell was dissolved by HCl to form $CaCl_2$ solution, adding NaOH to the solution dropwise to agitate Ca$(OH)_2$ gel and finally drying the gel at 900 °C for 1 h. The synthesized nanoparticle was characterized by scanning electron microscope (SEM), Fourier-transform infrared spectroscopy (FTIR), X-Ray fluorescence (XRF) and X-ray diffraction (XRD). The FTIR and XRD results have clearly depicted the synthesis of calcium oxide from eggshell, which is mainly composed of calcium carbonate. The FE-SEM images of calcium oxide nanoparticles showed that the particles were almost spherical in morphology. The particle size of the nanoparticles was in the range 50 nm–198 nm. Therefore, waste eggshell can be considered as a promising resource of calcium for application of versatile fields.

**Keywords:** calcium oxide nanoparticle; sol-gel method; eggshell

## 1. Introduction

Ever increasing solid waste is now becoming a challenge for a sustainable world. Improper management of those wastes leads to public health and environment related problems [1]. Huge amounts of solid wastes, including municipal, industrial and hazardous wastes, have been generated worldwide. Food wastes are the major solid wastes causing problems in the environment. It was estimated that food waste would increase by 44% from 2005 to 2025 [2]. Industrialization and population growth are the major factors for the increase in solid wastes. Eggshell is a solid waste which contributes to degradation in the environment. Households, restaurants, and bakeries are the major source of eggshell [3]. The main component is pure calcium carbonate with little porosity [3]. This waste can be transformed into valuable products. Eggshells can also be used as a $CaCO_3$ source for different applications [4,5]. However, the common trend to manage waste eggshell is land filling which causes problems for the environment. Landfilling such food wastes cause unpleasant odors

during biodegradation and the attached membranes attract worms and insects. Utilization of eggshells can provide benefits, not only regarding environmental concerns but also for freeing up landfill sites.

Synthesis of nanoparticles is attracting more attention because of better performance in terms of improved surface area. Metal oxide nanoparticles have many applications in diverse fields. They are considered as an active catalyst for versatile applications [6]. Metal oxide nanoparticles have been utilized as adsorbents for heavy metals removal in water and wastewater [7–11], as sorbents for $CO_2$ capture [12] where $CO_2$ capture increases with the increase in surface area of the particle [13], as heterogeneous catalysts in biodiesel production [14], as purification of exhaust gas [15] and in wall painting [16]. Calcium oxide, magnesium oxide, aluminum oxide, zinc oxide, manganese dioxide, titanium oxides, and iron oxide nanoparticles are the most commonly used nano-metal oxides. These metal oxide nanoparticles have been synthesized by several methods such as the ultrasonic-assisted method [17], the hydrogen plasma-metal reaction method [18], the biopolymer-assisted method [19] microwave-assisted method [20], facial calcination [21], co-precipitation [22], direct thermal decomposition [23], chemical co-precipitation [24], two-step process (green synthesis) [25] and two step thermal decomposition [26]. Those methods have drawbacks, such as the use of additives, high temperature, and pressure, time-consuming, expensive and complicated procedures. Sol-gel method overcomes most of the drawbacks from the above-mentioned methods. It is simple, cheap, not time consuming and no expensive equipment is required. It can also be carried out at lower temperature and with no pressure. Therefore, it can be a promising method to synthesize calcium oxide nanoparticles.

Nowadays, eggshell is being widely utilized for industrial applications [4,5,11,27]. Eggshell containing $CaCO_3$ as a major constituent can be a potential candidate for calcium oxide nanoparticle synthesis. In this study, CaO nanoparticles were synthesized by sol-gel method in which raw eggshell was dissolved by HCl to form $CaCl_2$ solution, adding NaOH to the solution dropwise to agitate $Ca(OH)_2$ gel and finally drying the gel at high temperature. This method can be considered as cheap, easy and eco-friendly. The objective of this work is to utilize waste eggshell in wastewater treatment systems.

## 2. Materials and Methods

### 2.1. Materials

Chemicals, hydrochloric acid with 35%–37% concentration and sodium hydroxide with 97% purity were purchased from Junsei Chemicals Ltd., Seoul, Korea. Waste chicken eggshells were collected from the KIGAM (Korean Institute of Geoscience and Mineral Resources) campus restaurant in Daejeon, South Korea. Raw waste eggshell is mainly composed of CaO, MgO, $K_2O$, $P_2O_5$, $Na_2O$ and some trace compounds.

### 2.2. Synthesis of Calcium Oxide Nanoparticles

Synthesis of metal oxide nanoparticle through sol-gel method has four basic consecutive processes: Preparation of homogeneous solution, the formation of 'sol' by hydrolysis, the formation of 'gel' by condensation and drying of the formed gel [28]. In this experiment, a homogenous solution of metal salt, $CaCl_2$, was produced by dissolving solid $CaCO_3$ in dilute HCl as shown in Equation (1). Chicken eggshell was used as a precursor for this method. Waste eggshells were thoroughly washed with warm water and cleaned with deionized water. Then the washed sample was dried at 120 °C for 2 h, ground into a powder and sieved with 100 μm sieve size. For the preparation of calcium chloride ($CaCl_2$) solution, 12.5 gm of PES was dissolved in 250 mL of 1 M hydrochloric acid (HCl).

$$CaCO_3 \text{ (s)} + 2HCl \text{ (aq)} \rightarrow CaCl_2 \text{ (aq)} + H_2O \text{ (l)} + CO_2 \text{ (g)} \tag{1}$$

The next step was the formation of 'sol' by hydrolysis process. 'Sol' is defined as a stable dispersion of colloidal particles of precursors in a solvent due to hydrolysis reaction. During the hydrolysis

process, metal hydroxide was formed. A total of 250 mL of 1 M sodium hydroxide (NaOH) was added slowly (drop by drop) to convert the homogeneous $CaCl_2$ solution formed in the previous step into 'sol' at ambient temperature (Equation (2)). Then 'gel' formation by condensation was followed. The slow addition of NaOH resulted in a low rate of nucleation and encouraged subsequent precipitation of $Ca(OH)_2$ one over another forming a highly crystalline gel. The condensation reaction resulted in small-sized particles interconnected to each other forming a rigid and highly crystalline inorganic network within the liquid. $Ca(OH)_2$ gel containing solution was aged for one night at ambient temperature.

$$CaCl_2 \text{ (aq)} + 2NaOH \text{ (aq)} \rightarrow Ca(OH)_2 \text{ (s)} + 2NaCl \text{ (aq)} \tag{2}$$

Then filtration followed where the filtrate was cleaned with distilled water in order to remove adsorbed impurities in the precipitate. The synthesizing process ended by drying the gel. In this process, the solvent (liquid phase) was removed and a significant amount of shrinkage and densification was noticed. The powder was dried at 60 °C for 24 h in an oven and calcined at 900 °C for 1 h (Equation (3)).

$$Ca(OH)_2 \text{ (s)} + \text{Heat} \rightarrow CaO \text{ (s)} + H_2O \text{ (l)} \tag{3}$$

### *2.3. Characterization*

Characterization of powders was determined by thermogravimetric analysis (TGA), X-ray fluorescence (XRF), Fourier-transform infrared spectroscopy (FTIR), X-ray diffraction (XRD) and scanning electron microscope (SEM). Decomposition temperatures of Raw eggshell and $Ca(OH)_2$ gel were analyzed by thermogravimetric analysis (TGA) (Shimadzu DTG-60H) in a platinum crucible at a heating rate of 10 °C/min from ambient temperature to 1000 °C. The chemical analysis of raw eggshell and synthesized CaO nanoparticle were performed using X-ray fluorescence spectrometer (Shimadzu, Japan). The crystal structural analysis was analyzed by X-ray diffraction (XRD) with diffraction angles 2θ from ranging 10° to 90°and with Cu Kα radiation (λ = 1.5406 Å) as the radiation source. Size, shape and surface morphology of the nanoparticle was examined through field emission scanning electron microscope (FE-SEM), (Tuscan Mira 3 LMU FEG) with a coater (Quorum Q150T ES/10 mA, 120 s Pt coating) at an accelerating voltage of 10 kV. The FTIR spectroscopy (FT-IR) (6700 FTIR, Thermo Scientific Nicolet, Massachusetts, MA, USA) was used to determine the different functional groups present in the synthesized nanoparticle.

### 3. Results

Thermogravimetric analysis (TGA) curves of raw eggshell and $Ca(OH)_2$ gel are illustrated in Figure 1a,b respectively. In raw eggshell, a total weight loss of 39% was noticed and at 823 °C temperature $CO_2$ is released to the environment. In the case of $Ca(OH)_2$ gel, three major weight losses were observed in the analysis: 32 °C–461.89 °C, 461.89 °C–691.2 °C and 691.2 °C–1000 °C with mass changes of 18.49%, 13.8% and 1.60% respectively. The losses correspond to vaporization of physically adsorbed water, decomposition of $Ca(OH)_2$ gel to CaO and decomposition of $CaCO_3$ to CaO where $CO_2$ will be released [23].

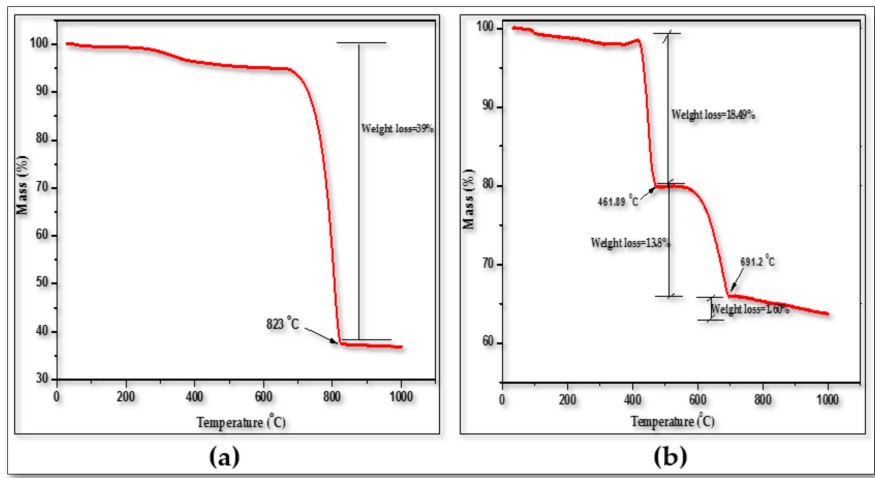

**Figure 1.** Thermogravimetric analysis (TGA) curve of (**a**) Raw eggshell, (**b**) Ca(OH)$_2$ gel.

The chemical composition of raw eggshell and synthesized nanoparticle is listed in Table 1. The data below shows that CaO is the major quantity in the raw eggshell, which is mainly in the form of CaCO$_3$ [29]. The High value of Ignition loss for raw eggshell represents the conversion of CaCO$_3$ to CaO and CO$_2$ during calcination [4]. After the synthesis of nanoparticle, lime was found to be the major component (86.93%).

**Table 1.** Chemical analysis of raw eggshell and nanoparticle.

| Chemical Composition | SiO$_2$ | Al$_2$O$_3$ | Fe$_2$O$_3$ | CaO | MgO | K$_2$O | Na$_2$O | TiO$_2$ | MnO | P$_2$O$_5$ | Ig. Loss |
|---|---|---|---|---|---|---|---|---|---|---|---|
| Raw eggshell (%) | <0.01 | <0.01 | <0.01 | 52.75 | 0.52 | 0.04 | 0.05 | <0.01 | <0.01 | 0.22 | 46.62 |
| Nano-CaO (%) | 0.08 | 0.04 | 0.05 | 86.93 | 1.08 | 0.14 | 1.32 | <0.01 | <0.01 | 0.43 | 9.3 |

FTIR spectrum of CaO nanoparticle, Ca(OH)$_2$ gel, raw eggshell and commercial CaO, Ca(OH)$_2$ and CaCO$_3$ are presented in Figure 2 to characterize the synthesized nanoparticle and also to compare the results with the commercial powders. The FTIR result of the raw eggshell showed broadband centering at 1415.52 cm$^{-1}$ which is a characteristic of C–O bond showing a bond between the oxygen atom of carbonate and calcium atom [30]. In addition, there were two sharp bands at 711.62 and 875.54 cm$^{-1}$ showing C–O bond [30]. The peaks of raw eggshell were in correspondence with commercial CaCO$_3$ except for the broadband at 2360.48 cm$^{-1}$ which represents the N–H bond caused by the amines and amides present in the protein fiber of the eggshell membrane [31]. On the other hand, CaO nanoparticles showed peaks at 1444.42 cm$^{-1}$, 1064.51 cm$^{-1}$, and 863.95 cm$^{-1}$ which were ascribed to C–O bond indicating the carbonation of calcium oxide nanoparticles [18,24]. The absorption peak at 3639.02 cm$^{-1}$ has also resulted due to the O–H bond from water molecules on the surface of the nanoparticle [18,24]. The tiny peak at 2343.09 cm$^{-1}$ might be due to atmospheric CO$_2$ [32]. This peak has also been seen in commercial CaO and Ca(OH)$_2$. The absence of a sharp absorption in the region at 1415.52 cm$^{-1}$ indicates that the CaCO$_3$ as the basic component of the eggshell was no longer present as it was already converted to CaO. The strong band at 512 cm$^{-1}$ identified vibration of the Ca–O band [20,24].

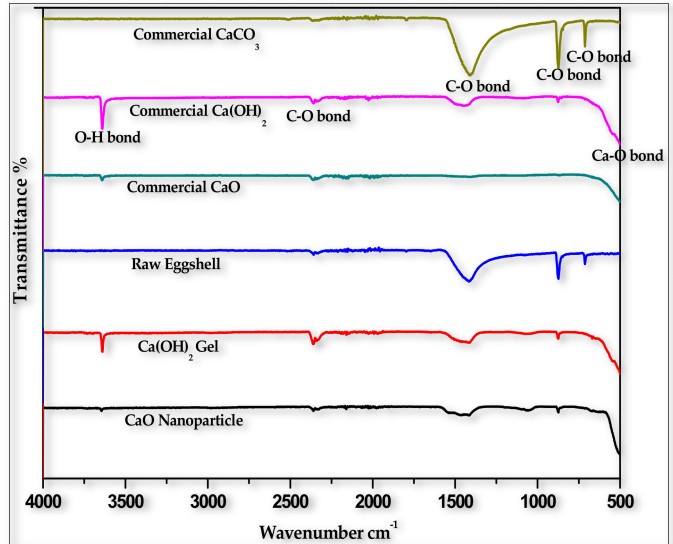

**Figure 2.** FTIR spectrum of CaO Nanoparticle, Ca(OH)$_2$ Gel, Raw eggshell and Commercial CaO, Ca(OH)$_2$ and CaCO$_3$.

XRD results of raw eggshell, commercial Ca(OH)$_2$, Ca(OH)$_2$ gel and synthesized calcium oxide nanoparticle are shown in Figure 3. Raw eggshell XRD pattern showed a good match with CaCO$_3$ in the calcite phase (PDF Card No. 00-081-2027). The main peak appeared at 2θ = 29.48. In addition, the analysis shows several peaks at 23.07, 31.52, 36.06, 39.48, 43.26, 47.64, 48.6, 56.62, 57.5, 60.92, 63.2, 64.76, 65.74, 70.29, 72.96, 76.37, 82.15, 84.9 which are assigned to (012), (006), (110), (113), (202), (018), (116), (211), (122), (214), (300), (0012), (0210), (128), (220), (114), (226) planes of calcite phase respectively. XRD patterns of Ca(OH)$_2$ gel match with portlandite phase (PDF Card No. 00-087-0673) as a major phase and also with commercial Ca(OH)$_2$ as shown the figure. The main peak of the gel appeared at 2θ = 34.12. In the case of synthesized calcium oxide nanoparticle (Figure 3), the XRD patterns match with calcium oxide (CaO) (PDF Card No. 99-0070). The main peak appeared at 2θ = 37.4. Besides, several peaks appeared at 32.22, 37.36, 53.9, 64.2, 67.4, 79.7 and 88.58 which are also assigned to (111), (200), (220), (311), (222), (400), and (331) planes of lime phase respectively. The result shows that the calcium carbonate in the raw eggshell was completely changed to calcium oxide during the synthesis.

Scherrer's Equation (Equation (4)) was used to calculate the mean crystallite size of the calcium oxide nanoparticle:

$$d = \frac{k\lambda}{\beta \cos \theta} \tag{4}$$

where d is mean crystallite size, λ is wavelength, k is constant of Scherrer, θ is Bragg's angle and β is structural broadening. Accordingly, the mean size of crystallite was found to be 24.51 nm. Other researchers using this technique of nanoparticle synthesis (sol-gel) have also confirmed the formation of large crystallite size. It was reported that the synthesis of MgO nanoparticles with an average particle size of 27.0 nm using the sol-gel technique [33]. Other studies also reported that the synthesized calcium oxide nanoparticle had a crystallite size of 40 nm and 41 nm for two different conditions [24]. Moreover, the lattice strain of crystal was determined by Equation (5):

$$\varepsilon = \frac{\beta}{4\tan \theta} \tag{5}$$

where θ is Bragg's angle, β is structural broadening, ε is lattice strain. Therefore, it was determined as $4.41 \times 10^{-3}$.

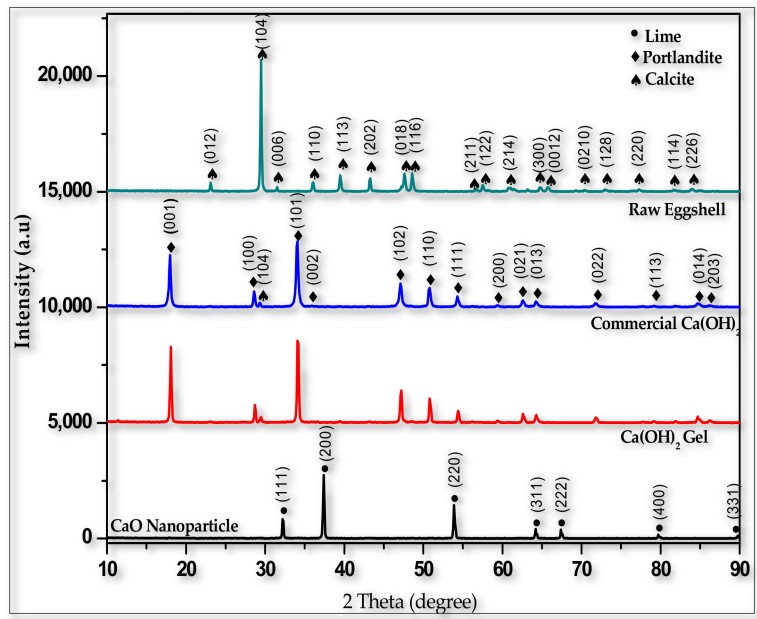

**Figure 3.** X-ray Diffraction of a Raw eggshell, Commercial Ca(OH)$_2$, Ca(OH)$_2$ Gel and CaO nanoparticle.

Surface morphology of Ca(OH)$_2$ gel and synthesized calcium oxide nanoparticle was studied by FE-SEM as shown in Figure 4ab. Raw eggshell had a non-porous and irregular crystal structure. In Figure 4a, the surface micro structural analysis, using SEM, of Ca(OH)$_2$ gel has been analyzed to study the transition from gel to nanoparticle obtained after drying. The result revealed that the Ca(OH)$_2$ gel has a hexagonal shape in which the size is also in nano-scale (around 450 nm). This result has also been obtained from other research of synthesizing Ca(OH)$_2$ nano-plates from oyster shells [34]. FE-SEM image shown in Figure 4b showed that the nanoparticles are approximately spherical in morphology agglomerating to each other. These agglomerates of small particles show the polycrystalline character of the nanoparticle. Other studies confirmed the spherical shape of calcium oxide nanoparticles [17,24]. Mean size of CaO nanoparticle was estimated to be 198 nm, as seen in Figure 5. Size of calcium oxide nanoparticle was reduced after drying the Ca(OH)$_2$ gel due to the release of CO$_2$ and H$_2$O. A calcination temperature of 900 °C was used, at which vaporization of absorbed water and decomposition of Ca(OH)$_2$ and CaCO$_3$ to CaO occurred. This result was also confirmed by XRD result where lime was obtained from the synthesis.

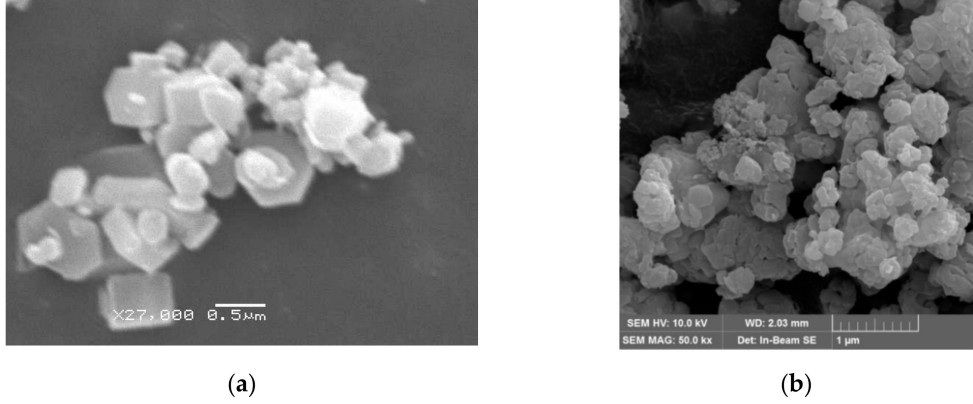

| (a) | (b) |

**Figure 4.** Scanning electron microscopy images of (**a**) Ca(OH)$_2$ gel, and (**b**) synthesized CaO nanoparticle.

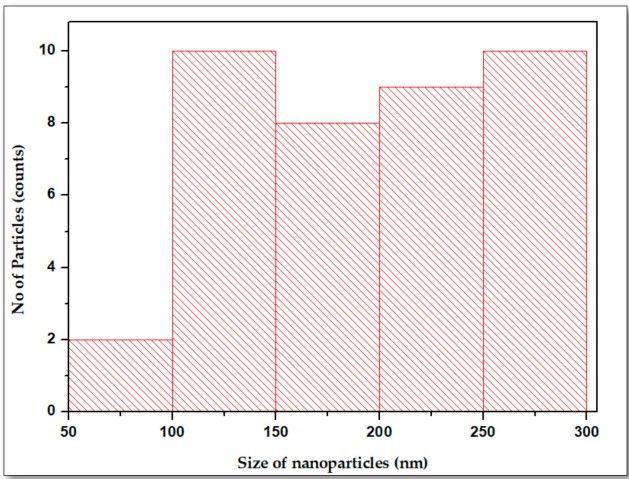

**Figure 5.** Particle size distribution of nano-calcium oxide.

Table 2 briefly summarizes comparison between current and other method of calcium oxide nanoparticle synthesis. Although particle size obtained in this technique is higher than other methods, it has advantages over the other methods. It is very simple, cheap, does not require expensive equipment, takes only a short time and has no additives. Moreover, in the synthesis process, ambient temperature was used, contributing to less energy consumption. Other methods used higher temperature, polymer additives, expensive equipment and took a longer time.

**Table 2.** Brief summary of CaO nanoparticles synthesis methods.

| Method | Summary | Reference |
|---|---|---|
| **Facial calcination** | A total of 50 nm of calcium oxide nano-particles were obtained by facile thermal treating of calcite at 900 °C temperature for 5 h and then by lime hydrolysis. | Malekzadeh et al., 2012 [21] |
| **Microwave irradiation** | By microwave processing, the calcium oxide nano-particles were obtained at 160 °C temperature in the 5 min with the average particle size is 14~24 nm. | Jayanta Bhattacharya et al., 2013 [20] |
| **Co-Precipitation** | CaO nano-particles are synthesized by the co precipitation method, in presence of polyvinylpyrrolidone (PVP) reagent for control the agglomeration. The average time takes for this synthesis was 12 h at 40 °C temperature. The average size of the nano-particles are 100 nm. | Meysam Sadeghi et al., 2013 [22] |
| **Direct thermal decomposition** | Calcium oxide nano-particles were synthesized by direct thermal decomposition method at 80 °C by blowing inert argon gas with the average particle size is 91 nm~94 nm. | Fereshteh Bakhtiari et al., 2014 [23] |
| **Chemical co-precipitation** | Chemical co-precipitation was applied for the synthesis of calcium oxide nano-particles in presence of polyvinyl alcohol. The average particle size is 11 nm at 80 °C for 60 min. | Ali et al., 2015 [24] |
| **Two step process (Green synthesis)** | CaO nano-particles were synthesized from shrimp cells by two step process with the average particle size is 40 to 130 nm. | Hui-Fen Wu et al., 2015 [25] |
| **Two step thermal decomposition** | Crystallite size of calcium oxide nano-particles were obtained by 2 step thermal decomposition method. | Arul et al., 2018 [26] |
| **Sol-gel method (present authors used)** | A total of 50–198 nm of calcium oxide nanoparticles was obtained at ambient temperature, with less cost, no additives, shorter time and a calcination temperature of 900 °C for 1 h only. | Ahn et al., 2019 |

## 4. Conclusions

In this paper, calcium oxide nanoparticles were synthesized from waste eggshell through the sol-gel method. Sol-gel technique has many key advantages over other methods for synthesizing metal oxide nanoparticles, such as being simple, economic, requiring no expensive equipment, ambient temperature and having no pressure. The FTIR and XRD results have clearly depicted the synthesis of calcium oxide from eggshell, which was mainly composed of calcium carbonate. The FE-SEM images of Calcium Oxide nanoparticles showed that the particles were almost spherical in morphology. The mean particle size of the nanoparticles was 198 nm. But this result can be improved if higher temperature is used. The synthesized nanoparticle can be applied for future studies in heavy metal removal from industrial wastewater. Moreover, utilizing waste materials as a precursor for the synthesis makes the whole process cheaper, green and sustainable. Waste eggshell can further be used in future work for the synthesis of nano-calcium carbonate, which can be applied as a filler material in the automobile and paper industry.

**Author Contributions:** L.H., N.S., D.M., conducted the experiments and wrote the manuscript. T.T., R.C. collected the information, analyzed the data and A.J.W. corrected the final manuscript and agreed to submit this data to the sustainability journal.

**Funding:** This research was funded by Ministry of Science and ICT(MSIT), the Ministry of Environment (ME) and the Ministry of Trade, Industry, and Energy (MOTIE) and the Grant number [2017M3D8A2084752] and the APC National Strategic Project-Carbon Mineralization Flagship Center of the National Research Foundation of Korea (NRF).

**Acknowledgments:** This research was supported by the National Strategic Project-Carbon Mineralization Flagship Center of the National Research Foundation of Korea (NRF) funded by the Ministry of Science and ICT(MSIT), the Ministry of Environment (ME) and the Ministry of Trade, Industry, and Energy (MOTIE). (2017M3D8A2084752).

**Conflicts of Interest:** The authors declare no conflict of interest.

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
