# Peer review of "Synthesis of Nano-Calcium Oxide from Waste Eggshell by Sol-Gel Method"

_sustainability, doi:10.3390/su11113196_

Reviewer 1 Report

The paper describes utilization of egg-shells for the production of nano-particles using sol-gel method. Used procedures are clearly described and used English is understandable. However, egg-shells are natural material, hence there should be clearly explained environmental risks of their land filling in the introduction.

Author Response

1st reviewer comments

The paper describes utilization of egg-shells for the production of nano-particles using sol-gel method. Used procedures are clearly described and used English is understandable. However, egg-shells are natural material, hence there should be clearly explained environmental risks of their land filling in the introduction.

Ans) Thank you for your comments. Yes sir we added the environmental risks of egg shells landfilling in the revised manuscript.

Reviewer 2 Report

The manuscript titled: "Synthesis of Nano-Calcium oxide from Waste Eggshell by Sol-gel Method" written by L. Habte and co-workers deals with the conversion of chicken eggshells (calcium carbonate) to CaO nanoparticles via sol-gel methods (intermediate are CaCl2 and Ca(OH)2). The study here proposed is interesting and very promising for future applications. However, major revision is necessary. In details:

1) Several typos were present in the entire document. Please, Authors should correct them. Also the English style can be improved.

2) FTIR spectra should be put in the same graph to better compare signals' modifications. Additionally, a reference spectra of commerciale CaCO3 and CaO should be included and compared with the experimental ones. Authors should include also the spectra of Ca(OH)2 gel.

3) In order to follow the CO2 and CO evolution, TGA analysis should be added.

4) XRD analysis. Firstly, XRD is NOT a spectroscopy. Please, include also the XRD of Ca(OH)2 vela and reference pattern in the same graph.

5) This Reviewer does dot understand why NaCl signals are not present. Authours could clarify why the XRD analysis revealed only CaO ones?

6) The paragraph from "Synthesis of metal nanoparticles" to "noticed" should be rewritten comparing the results here obtained with the state of art literature: why this manuscript should be published?

7) Within the entire article and abstract Authors wrote "metal nanoparticles" but CaO is a metal oxide nanoparticles! Please correct it.

8) Which is the amount of residual protein in CaO NPs?

Author Response

Answers to the 2nd reviewer comments

The manuscript titled: "Synthesis of Nano-Calcium oxide from Waste Eggshell by Sol-gel Method" written by L. Habte and co-workers deals with the conversion of chicken eggshells (calcium carbonate) to CaO nanoparticles via sol-gel methods (intermediate are CaCl2 and Ca(OH)2). The study here proposed is interesting and very promising for future applications. However, major revision is necessary.

Ans) Thank you very much for comments and valuable suggestions.

In details:

1) Several typos were present in the entire document. Please, Authors should correct them. Also the English style can be improved.

Ans) Yes sir. We carefully checked the typos in the entire manuscript and revised it.

2) FTIR spectra should be put in the same graph to better compare signals' modifications. Additionally, a reference spectra of commerciale CaCO3 and CaO should be included and compared with the experimental ones. Authors should include also the spectra of Ca(OH)2 gel.

Ans) Yes we added the FT IR spectra in the revised manuscript.

3) In order to follow the CO2 and CO evolution, TGA analysis should be added.

Ans) Yes we added the TGA analysis data in the revised manuscript

4) XRD analysis. Firstly, XRD is NOT a spectroscopy. Please, include also the XRD of Ca(OH)2vela and reference pattern in the same graph.

Ans) Yes we added the XRD of Ca(OH)2 gel

5) This Reviewer does dot understand why NaCl signals are not present. Authors could clarify why the XRD analysis revealed only CaO ones?

Ans) NaCl signals are not present in the XRD analysis because it was filtered out in the solution and present in aqueous form in the solution. As we have mentioned in the methodology, Ca(OH)2 gel was formed after the addition of NaOH. Then the gel was filtered out from the solution and dried for analysis. Therefore, NaCl is left in the filtered solution in dissolved form.

6) The paragraph from "Synthesis of metal nanoparticles" to "noticed" should be rewritten comparing the results here obtained with the state of art literature: why this manuscript should be published?

Ans) Yes sir. we added the information in the revised manuscript.

7) Within the entire article and abstract Authors wrote "metal nanoparticles" but CaO is a metal oxide nanoparticles! Please correct it.

Ans) Yes sir. We modified the word in the revised manuscript.

8) Which is the amount of residual protein in CaO NPs?

Ans) Thank you for the valuable question.

Initially, we decided that the peak that appeared in the CaO  NPs at 2343.09 cm-1 was still due to the amine groups present in the protein of eggshell membrane. And we know that some bonds have almost similar peaks. But actually when we conduct the FT IR of commercial CaO and Ca(OH)2 , we noticed the same peak around 2500 cm-1. In addition while calcination all organic matter will disappear. Therefore, we have amended the interpretation in the revised manuscript. Instead of N-H bond due to amines, it might be due to atmospheric CO2, resulting in C-O bond.

Reviewer 3 Report

The proposed method serves the utilization of waste as well as the acquisition of valuable chemical compounds. The work which is done by the authors is important for science, industry and for protection of environment. Physico-chemical analyzes - which were performed - are adequate and their interpretation is correct.

I recommend publishing this article, but after some corrections.

Comments:
- Generally, in this paper, both in the Introduction and in Results, there is a lack of good discussion, i.e. references to adequate results contained in the literature. Please extend the description about the advantage of your method, please. The one presented at work – it’s too laconic.

- How were the results presented in Table 1 obtained? I don’t notice the explanation in the "Methodology". Complete constantly the data in the "Methodology" (technique, names of all apparatus used, company etc.), please.

- I have not found the analysis of the figure 4. The descriptions of the axes on the graph 4 – are too laconic - please change this (eg ‘size’, but - whose?; ‘counts’… ?)

- The last paragraph of Chapter 3 is, in my opinion, detached from the whole. Link it with the results obtained and complete with the discussion, please.

Author Response

Answers to the 3rd reviewer comments

1.The proposed method serves the utilization of waste as well as the acquisition of valuable chemical compounds. The work which is done by the authors is important for science, industry and for protection of environment. Physico-chemical analyzes - which were performed - are adequate and their interpretation is correct. 

Ans) Thank you very much sir.

I recommend publishing this article, but after some corrections.

Comments:

2. Generally, in this paper, both in the Introduction and in Results, there is a lack of good discussion, i.e. references to adequate results contained in the literature. Please extend the description about the advantage of your method, please. The one presented at work – it’s too laconic.

Ans) Yes sir. We added more relevant references and description in the revised manuscript in the line no.s…

3.How were the results presented in Table 1 obtained? I don’t notice the explanation in the "Methodology". Complete constantly the data in the "Methodology" (technique, names of all apparatus used, company etc.), please.

Ans) Yes sir. We added all the details in the methodology section in the revised manuscript in the line no.s…

4.I have not found the analysis of the figure 4. The descriptions of the axes on the graph 4 – are too laconic - please change this (eg ‘size’, but - whose?; ‘counts’… ?)

Ans) Yes sir. We modified the figure 4 in the revised manuscript.

 5. The last paragraph of Chapter 3 is, in my opinion, detached from the whole. Link it with the results obtained and complete with the discussion, please.

Ans) Yes sir. We modified the section 3 with the results and discussion in the revised manuscript in the line no…..

Round  2

Reviewer 2 Report

The present version can be accepted for publication.